# Empowering consumers to PREVENT diet-related diseases through OMICS sciences (PREVENTOMICS): protocol for a parallel double-blinded randomised intervention trial to investigate biomarker-based nutrition plans for weight loss

Mona Adnan Aldubayan [1,2] Kristina Pigsborg [1] Sophia M O Gormsen,[3] Francisca Serra,[4,5] Mariona Palou,[4,5] Pedro Mena,[6] Mart Wetzels,[7] Alberto Calleja,[8] Antoni Caimari,[9] Josep Del Bas,[9] Biotza Gutierrez,[9] Faidon Magkos,[1] Mads Fiil Hjorth[10]

For numbered affiliations see end of article.

**Correspondence to**
Mona Adnan Aldubayan;
monal@nexs.ku.dk

## ABSTRACT

**Introduction** Personalised nutrition holds immense potential over conventional one-size-fits-all approaches for preventing and treating diet-related diseases, such as obesity. The current study aims to examine whether a personalised nutritional plan produces more favourable health outcomes than a standard approach based on general dietary recommendations in subjects with overweight or obesity and elevated waist circumference.

**Methods and analysis** This project is a 10-week parallel, double-blinded randomised intervention trial. We plan to include 100 adults aged 18–65 years interested in losing weight, with body mass index ≥27 but<40 kg/m$^2$ and elevated waist circumference (males >94 cm; females >80 cm). Participants will be categorised into one of five predefined 'clusters' based on their individual metabolic biomarker profile and genetic background, and will be randomised in a 1:1 ratio to one of two groups: (1) personalised plan group that will receive cluster-specific meals every day for 6 days a week, in conjunction with a personalised behavioural change programme via electronic push notifications; or (2) control group that will receive meals following the general dietary recommendations in conjunction with generic health behaviour prompts. The primary outcome is the difference between groups (personalised vs control) in the change in fat mass from baseline. Secondary outcomes include changes in weight and body composition, fasting blood glucose and insulin, lipid profile, adipokines, inflammatory biomarkers, and blood pressure. Other outcomes involve measures of physical activity and sleep patterns, health-related quality of life, dietary intake, eating behaviour, and biomarkers of food intake. The effect of the intervention on the primary outcome will be analysed by means of linear mixed models.

### Strengths and limitations of this study

► This study may identify novel approaches in facilitating weight loss and health-promoting behaviours by applying state-of-the-art knowledge that integrates metabolomics and genetics with nutrition.

► The trial is double-blinded, which is rare in nutritional science, and serves as proof of concept for the personalised dietary management of obesity.

► A potential limitation is that both groups are receiving healthy foods and behavioural advice, which may mask the hypothesised intervention effect of the personalised plan.

► The study is powered to detect differences in 10-week body fat loss between intervention and control arms and not within each of the five clusters; differences between the latter will be assessed by post hoc analysis.

► Potential long-term effects of a personalised approach cannot be evaluated from this 10-week study; however, results will provide a basis for implementation in longer obesity-management programmes.

**Ethics and dissemination** The protocol has been approved by the Ethics Committee of the Capital Region, Copenhagen, Denmark. Study findings will be disseminated through peer-reviewed publications, conference presentations and media outlets.
**Trial registration number** NCT04590989.

## INTRODUCTION

The ultimate goal of nutrition research and dietary recommendations is the

promotion of human health and the prevention or treatment of chronic diseases.[1] Still, the global prevalence rate of nutrition-related non-communicable diseases (NCDs) continues to rise rapidly.[2] There is considerable evidence indicating that obesity is a major risk factor for developing NCDs including type 2 diabetes, cardiovascular diseases and certain types of cancers, which are the leading causes of morbidity and mortality.[3 4] Therefore, obesity puts a great burden on the individual, the healthcare system and society.[3] Accordingly, enormous efforts to tackle this epidemic have been implemented from health professionals through setting different policies and guidelines for the public, but with little success, as management of obesity remains a very challenging task. Moreover, the optimal diet characteristics—particularly with respect to dietary macronutrient composition (eg, low-carbohydrate, low-fat, high-protein diets)—that are most effective in reducing excess weight gain or promoting weight loss have long been debated.[5 6] Clinical trials have demonstrated that certain individuals benefit more from a particular dietary intervention than others in reducing body weight, while only a small number are able to keep the weight off in the long term.[7 8] This implies there is no strong evidence that one diet is superior to others for inducing weight loss, and there is no such thing as a 'perfect' diet for everyone. Such substantial interindividual variation in response to any given dietary treatment can be attributed to multiple phenotypic factors and genetic variants which influence how the body utilises and metabolises nutrients.[7] This gives rise to the demand for customising diet plans and nutrition advice at the individual or small group level, rather than at the population level. Recent developments in 'omics' technologies (nutrigenomics, transcriptomics, epigenomics, metabolomics, metagenomics) offer exciting opportunities to explore the complex interplay between nutrition, genetics and metabolism.[9] By integrating these novel tools with bioinformatics, the potential of 'personalised nutrition' can be implemented through identifying novel biomarkers that can predict the most effective diet for weight loss and improved health outcomes for any given individual.[9–11] Therefore, the ability to provide evidence-based dietary advice based on individual genetic make-up, phenotypic information on anthropometry, biochemical and metabolic profiles, physical activity habits and medical history—among others—may lead to changed behaviours and ultimately, improved health.

In this context, H2020 PREVENTOMICS (Empowering consumers to PREVENT diet-related diseases through OMICS sciences), coordinated by Eurecat in Spain, has developed a platform with a Decision Support System (DSS) tool that integrates individual phenotypic characteristics at the metabolome level with their genotype, lifestyle habits, and preferences to improve their health status through personalised nutrition management plans. The project aims to examine the validity of the PREVENTOMICS platform in terms of its potential for personalisation at different levels of the food value chain. This will be achieved through different intervention studies in Denmark, Spain, and Poland and the UK, with both healthy volunteers and volunteers with abdominal obesity. Here we report the specific characteristics of the Danish study protocol.

## Research hypothesis and aims

The overall aim of this 10-week randomised trial is to examine the efficacy of the PREVENTOMICS platform, integrated in an e-commerce digital tool created for delivering personalised meals for producing more favourable health outcomes than meals based on general dietary recommendations, in subjects with overweight or obesity and elevated waist circumference. In addition, the intervention group will receive tailored and actionable behaviour change prompts whereas the control group will receive general nutrition and lifestyle advice. Our hypothesis is that the personalised dietary and behavioural treatment plan will produce greater reductions in fat mass and body

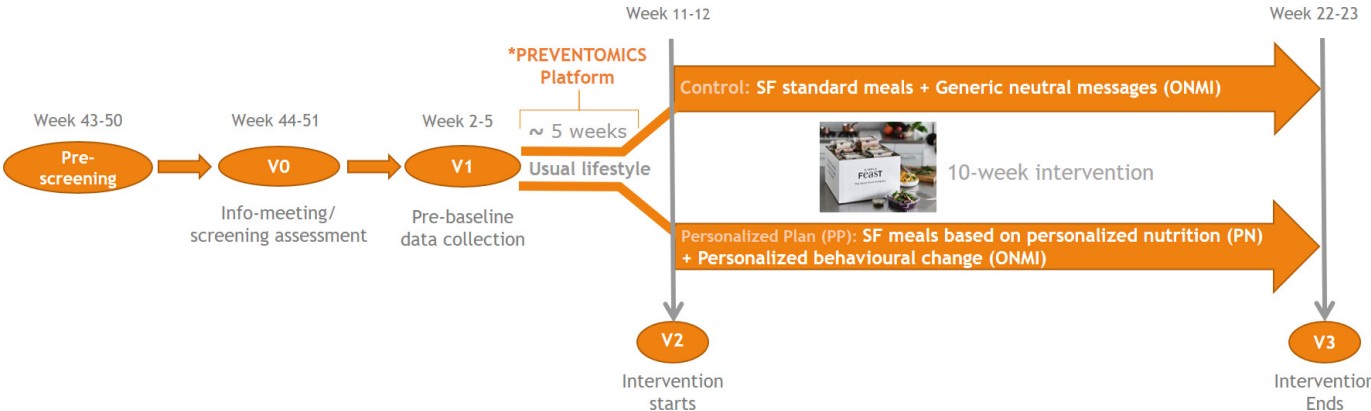

**Figure 1** Study design and timeline. *The results for metabolome and genotype analyses are integrated into the PREVENTOMICS platform. PREVENTOMICS, Empowering consumers to PREVENT diet-related diseases through OMICS sciences; SF, simple feast.

weight, and will promote more favourable changes in circulating metabolic and inflammatory biomarkers compared with the control dietary and behavioural treatment plan.

## Primary and secondary objectives

The primary goal of this study is to evaluate the change in body fat mass between the personalised plan group and the control group over the 10-week intervention period. The secondary goal is to evaluate the change in the following health outcomes between the personalised and control groups: (1) body composition (visceral and subcutaneous fat, lean body mass, weight, body mass index (BMI), waist circumference); (2) lipid profile (total cholesterol, low-density lipoprotein (LDL), high-density lipoprotein (HDL), triglycerides); (3) glucose homoeostasis (glucose, insulin, homoeostatic model assessment of insulin resistance (HOMA-IR)); (4) inflammatory markers (C reactive protein (CRP), interleukin-6 (IL-6), interleukin-10 (IL-10), monocyte chemoattractant protein-1 (MCP-1), tumour necrosis factor alpha (TNF-α), soluble amino acid residue glycosylated peptide-1 (ICAM-1), soluble cluster of differentiation-14 (CD-14), oxidised LDL (oxLDL)); (5) adipokines (leptin,

adiponectin); (6) liver function markers (alanine transaminase (ALT), gamma-glutamyl transferase (GGT)); (7) renal function markers (uric acid, creatinine) and (8) blood pressure.

## METHODS AND ANALYSIS
### Study design

This is a randomised, single-centre, parallel-group (1:1 ratio), double-blinded intervention study conducted at the research facilities of the Department of Nutrition, Exercise and Sports (NEXS), University of Copenhagen, Denmark. The study protocol adheres to the Standard Protocol Items: Recommendations for Interventional Trials guidelines.[12] Recruitment for this study started at NEXS immediately after trial registration. Information needed to determine the 'cluster' of participants was collected in January 2021, and the actual analysis of biological samples and clustering was performed in February–March 2021. All data collected during the 10-week intervention period (March–June 2021) are expected to be fully analysed by the end of 2021. The overall study design is illustrated in figure 1.

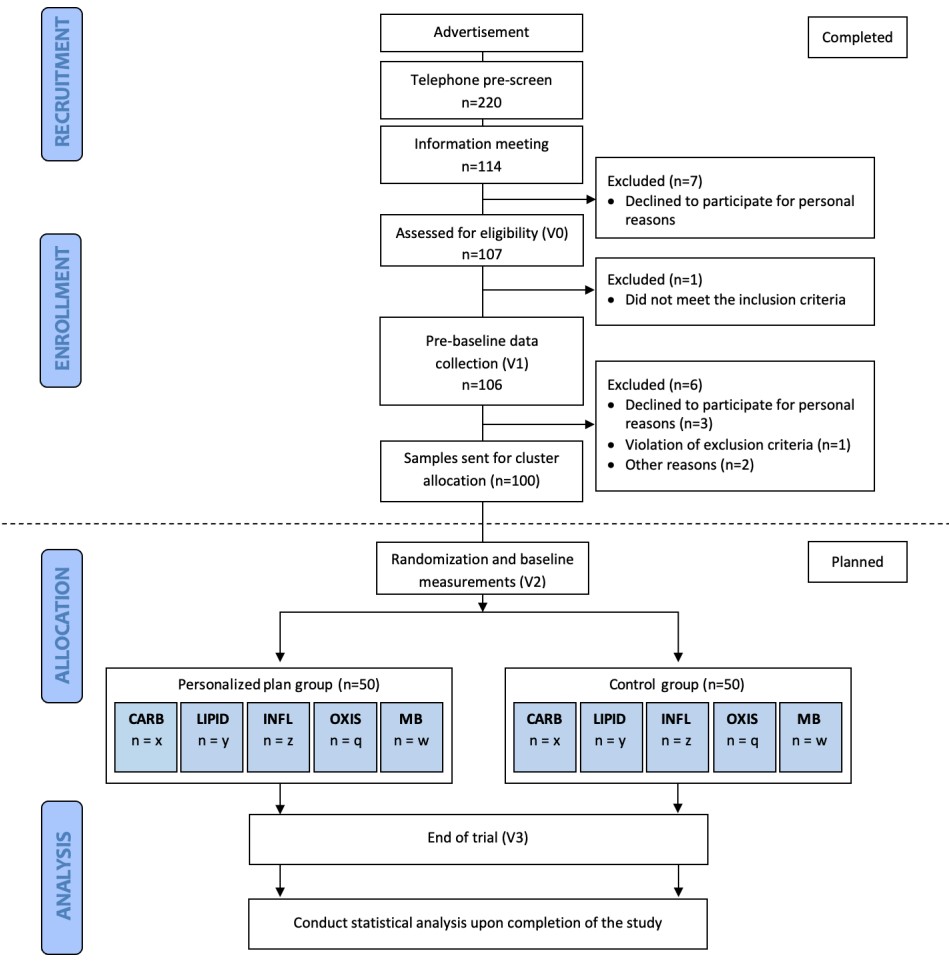

**Figure 2** Schematic diagram of the intervention. CARB, carbohydrate cluster; INFL, inflammation cluster; LIPID, lipid cluster; MB, microbiota cluster; OXIS, oxidative stress cluster.

### Patient and public involvement

Patients and the public were not involved in the design, conduct or reporting of this study.

### Study participants

Participants are males and females aged 18–65 years with a BMI of ≥27 but <40 kg/m$^2$ and elevated waist circumference (males >94 cm; females >80 cm). Participants should possess a smartphone and be able to provide an informed consent. The exclusion criteria are as follows: (1) diagnosis of diabetes; (2) history or diagnosis of heart, liver or kidney diseases; (3) chronic diseases, for example, cancer within the past 5 years (except adequately-treated localised basal cell skin cancer); (4) use of drugs (eg, antibiotics) that, in the opinion of the medically responsible investigator, are likely to affect the primary outcomes of the study; (5) being lactating, pregnant or planning to become pregnant within the study period; (6) self-reported weight change of >5% within 2 months prior to screening; (7) participation in another clinical trial; (8) other blood donation during the study; (9) having allergies or food intolerances; (10) no or limited access to the internet. Participants unable to comply with the study protocol, as judged by the investigator, are also excluded.

### Recruitment procedure

The study flow chart is summarised in figure 2. Potential participants were recruited through internet-based advertisements. Trained study personnel contacted 220 subjects who expressed interest in the study via telephone to determine initial eligibility (prescreening). Written information about the study was provided to 120 potential participants who were deemed eligible from the telephone pre-screening and scheduled for an oral information meeting (visit 0, V0) at the department (NEXS). If the subject signed the informed consent, either immediately following the information meeting or after a few days of consideration, they were screened according to the inclusion/exclusion criteria to assess final eligibility. A total of 106 participants were recruited and invited for the prebaseline visit (V1) where anthropometric measurements, blood, saliva and urine samples were collected, and various questionnaires were filled out. One hundred participants completed V1 and had their samples sent to the assigned consortium for analysing data on subjects' metabolome and genotype in addition to lifestyle habits, preferences and physiological status. These data are utilised to determine subjects' cluster (see later) and develop the personalised dietary plans for the subsequent 10-week intervention period.

### Cluster allocation

All subjects were categorised into one of five predefined 'clusters' (table 1) based on their metabolic and genetic biomarkers collected at V1, according to the following procedure:

**Table 1** Full list of biomarkers in relation to the metabolic clusters

| Carbohydrate | Lipid | Inflammation | Oxidative stress | Microbiota |
|---|---|---|---|---|
| Glucose | LDL-cholesterol | CRP | 8-iso-PGF2α | TMA |
| HOMA-IR | Total cholesterol | N-acetylglycoproteins | 8-OHdG | TMAO |
| Glutamate | PUFAs | MCP-1 | Oxidised LDL | Betaine |
| Uric acid | HDL-cholesterol | TNF-α | Uric acid | Choline |
| Leptin | SFAs | IL-6 | Allantoin | DMA |
| Adiponectin | Triglycerides | IL-10 | Betaine | Dimethylglycine |
| Insulin | MUFAs | SFAs | Pseudouridine | LBP |
| Tyrosine | LPC | sICAM-1 | Dimethylglycine | Succinate |
| Propionylcarnitine | Linoleic acid | LPC | Methionine | Lactate |
| Lactate | DHA | LBP | Glycine | Acetate |
| Valine | Oleic acid | DHA C20:3 | | |
| Leucine | Choline | sCD-14 | | |
| Isoleucine | 3-hydroxybutyrate | Linoleic acid C18:2 | | |
| Phenylalanine | Propionylcarnitine | PUFAs | | |
| Glutamine | Adiponectin | | | |
| | Leptin | | | |

Some biomarkers help define more than one cluster.

CRP, C reactive protein; DHA, docosahexaenoic acid; DMA, dimethylamine; HDL, high-density lipoprotein; HOMA-IR, homoeostatic model assessment of insulin resistance; IL-6, interleukin-6; IL-10, interleukin-10; 8-iso-PGF2α, prostaglandin, 8-iso-prostaglandin F2α; LBP, lipopolysaccharide binding protein; LDL, low-density lipoprotein; LPC, lysophosphatidylcholine; MCP-1, monocyte chemoattractant protein-1; MUFAs, monounsaturated fatty acids; 8-OHdG, 8-hydroxydeoxyguanosine; PUFAs, polyunsaturated fatty acids; sCD-14, soluble cluster of differentiation-14; SFAs, saturated fatty acids; sICAM-1, soluble amino acid residue glycosylated peptide-1; TMA, trimethylamine; TMAO, trimethylamine N-oxide; TNF-α, tumour necrosis factor alpha.

**Table 2** List of SNPs in relation to the metabolic clusters

| Lipid | | Carbohydrate | | Oxidative | | Inflammation | |
|---|---|---|---|---|---|---|---|
| Gene | SNP | Gene | SNP | Gene | SNP | Gene | SNP |
| ADIPOQ | rs182052 | ADIPOQ | rs182052 | COMT | rs4680 | APOE | rs429358 |
| APOA5 | rs12272004 | ASCL1 | rs17450122 | CPS1 | rs1047891 | CADM3-AS1 | rs12075 |
| APOA5 | rs662799 | FADS1, FADS2 | rs174550 | CPS1 | rs715 | CUX1 | rs409224 |
| APOE | rs7412 | GCKR | rs1260326 | FGF21 | rs838133 | FADS1 | rs174547 |
| APOE | rs429358 | GCKR | rs780093 | GSTP1 | rs1695 | GCKR | rs1260326 |
| CUX1 | rs409224 | GLS2, SPRYD4 | rs2657879 | MTHFR | rs1801133 | GCKR | rs780093 |
| FADS1 | rs174547 | LEP | rs10487505 | SOD2 | rs4880 | ICAM1 | rs5498 |
| GCKR | rs780093 | PPARG | rs1801282 | | | IL-6 | rs1800795 |
| GCKR | rs1260326 | SLC16A10 | rs14399 | | | | |
| HFE | rs1800562 | SLC16A9 | rs1171614 | | | | |
| LEP | rs10487505 | SLC2A2 | rs8192675 | | | | |
| LPL | rs268 | TCF7L2 | rs7903146 | | | | |
| LPL | rs326 | | | | | | |
| PNPLA3 | rs738409 | | | | | | |
| PPID | rs8396 | | | | | | |
| SLC16A9 | rs1171614 | | | | | | |
| TIMP3 | rs12678919 | | | | | | |
| TRIM58 | rs3811444 | | | | | | |

SNP, single-nucleotide polymorphism.

First, collected samples of urine, plasma and serum were analysed to assess a total of 51 biomarkers relevant to the following five metabolic processes: (1) oxidative stress; (2) inflammation; (3) carbohydrate metabolism; (4) lipid metabolism; and (5) gut microbiota metabolism. References supporting the rationale for using the biomarkers listed in table 1 are included in online supplemental material 1.

Second, a core of 35 different single nucleotide polymorphisms (SNPs) which are associated with the five metabolic processes and are able to modulate the biomarker levels reported in table 1 have been identified by Alimentomica (Spain) to be analysed in saliva samples (table 2). The biomarkers of the lipid cluster are able to be modulated, at different degree, by a set of 18 SNPs in 14 genes,[13–20] the carbohydrate cluster is represented by a panel of 12 SNPs in 11 genes[14 15 21–24] and the inflammation cluster by 8 SNPs in 7 genes.[25–28] Parse scientific data deal with the genetic impact on the specific biomarkers of the oxidative stress cluster; the corresponding genetic risk score comprises 7 genetic variants in 6 genes associated with reduced ability to buffer the oxidative stress associated with low levels of plasma antioxidants.[29–33] In relation to the microbiota cluster, the current evidence base does not provide enough data in support of the role of SNPs (included in the panel or not) and some evidence concerning the host genetic influence on microbiota response and on microbiota metabolite production is neither robust nor sufficient; hence no genetic influence was adopted in this cluster.

Finally, the specific SNPs and the biomarkers in the five metabolic processes will be used—by means of a proprietary algorithm—to calculate individual scores for each of the five metabolic clusters for any given participant. Each subject is then assigned to the metabolic cluster with the highest score. Briefly, the individual biomarkers, both metabolic and genetic, are combined into metabolic clusters considering both the absolute value of the biomarker in the biofluid and the biological relevance of the biomarker within the metabolic cluster. Whereas the first value is directly obtained from blood and urine measurements (metabolomics and proteomics biomarkers) and saliva (genotyping), the second is obtained from different approaches combining artificial intelligence applied to measurements of different biobank samples and literature review.[34] Therefore, the resulting score for each cluster is not based on the definition of thresholds but on the contribution of all individual biomarkers analysed. The specifics of the algorithm cannot be disclosed due to a pending intellectual property rights application.

### Randomisation and concealment

Prior to the intervention, all participants are stratified by metabolic cluster (oxidative stress; inflammation; carbohydrate metabolism; lipid metabolism; microbiota-generated metabolites) and then randomly assigned to either the control or the intervention group, in a 1:1 allocation ratio, by using a computer-generated randomisation sequence with random permuted block sizes of two subjects within each stratum. The person responsible for

randomisation and generating the code does not take part in the inclusion and examination of study participants.

In order to maintain blinding, the clustering results for all participants are shared by Eurecat before the baseline visit (V2) with a member of the staff at NEXS (not through the platform) who is only responsible for randomisation. Consequently, the allocation information is incorporated into each participant's profile in the PREVENTOMICS platform after passing back the randomisation information to Eurecat by the staff member at NEXS. However, this field is visible only to SimpleFeast (Denmark) and ONMI (The Netherlands; behavioural change technology, https://www.onmi.design/) user accounts for appropriate delivery of food and behavioural prompts. The profile includes the assigned group (personalised or control) and cluster, in addition to relevant recommendations to each subject. The local team of investigators at NEXS as well as participants are therefore unaware of the clustering and the randomisation of the participants as this field is hidden from their user account. Moreover, the statistical analyses of the main outcome variable will be conducted without breaking the code for the intervention treatment until the primary analyses have been finalised.

### Interventions
#### 1. Dietary intervention
During the 10-week intervention, the personalised plan and control groups receive easy-to-prepare meal boxes twice a week from Simple Feast (Copenhagen, Denmark) complying with the national dietary guidelines of macronutrient distributions.[35] Each delivery provides meal boxes of breakfast and dinner for the subsequent 3 days (12 meals/week). Meal boxes for the two groups are designed to be visually identical. Moreover, food by Simple Feast is vegetarian and organically produced, however, participants are allowed to eat non-organic/non-vegetarian foods as part of the meals not provided. The number of meals provided to the participants was decided on a combination of factors including budgetary limitations, practical reasons, and behavioural factors. For the days for which meals are not provided (Saturdays), as well as for all lunches, participants are encouraged to refer to the recipe recommendations that are presented through the Simple Feast Recipe App, so they prepare meals as similar as possible to the group and cluster they are assigned to. All provided foods, including the recipes, are plant-based and align with the recommended list of foods created by Eurecat for each group/cluster (see below).

The calorie content of meals was calculated based on the average daily energy requirements for the general population, which is 2000 kcal/day for females and 2500 kcal/day for males.[36] Given that 25% of daily energy is commonly consumed at breakfast and 35% at dinner, breakfast was designed to provide approximately 500 and 625 kcal/day (for females and males, respectively) and dinner to provide approximately 700 and 875 kcal/day (for females and males, respectively). Participants are instructed to consume the provided meals in whole, or until they are fully satisfied. In addition, they are advised to be inspired by the breakfast and dinner meals provided during this 10-week trial and to consume similar foods for lunch and limit intake of energy-dense foods and drinks. Plant-based meals are fibre-rich and induce greater and faster satiety.[37–39] We thus anticipate lower food consumption both for the provided meals but also outside them (ie, during lunch or when snacking), which will hopefully be large enough to produce the calorie deficit needed for body weight and fat loss. It is also anticipated that the personalised plan induces favourable changes in eating behaviour and physiological and metabolic parameters that promote body weight and fat loss when compared with the control plan.

The Eurecat Nutrition Team has prepared a list of recommended food items to increase, decrease, or completely exclude from the diet for the control and each cluster in the personalised group. The list was adopted by Simple Feast in creating five different menus that encompass 12 meals/week for the five different clusters in the intervention group in addition to the menu for the control group. Contrary to the control meals, personalised meals also include some bioactive compounds provided by CARINSA (Spain). These bioactive compounds were selected following review of the literature and are believed to benefit especially—or exclusively—the metabolic function of individuals in the corresponding cluster (table 3). Each participant received approximately 20 g of functional ingredient per day, except for the inflammation cluster (6–8 g per day). The macronutrient distribution of the diets between clusters differed only in the amount of fibre. Dietary fibre content was higher in the Carbohydrate and Microbiota clusters as these clusters received fructooligosaccharide and inulin as functional ingredients. Nutritional information on macronutrient content and the bioactive ingredients of the meals for each cluster in the personalised and control groups, as well as an example of a 3-day menu, are provided in online supplemental material 2.

#### 2. Behavioural assessment and intervention
All participants are asked to fill out a behavioural questionnaire at baseline (V2), in order to collect information about certain habits or behaviours that affect physical, emotional, or mental well-being. During the 10-week intervention period, both groups are enrolled in a behavioural programme delivered through ONMI's App with 2–3 electronic push notifications per week. Subjects randomised to the personalised group receive behavioural prompts (active Do's) from the predefined ONMI's evidence-based behavioural change programme, which has been developed to increase behavioural flexibility and facilitate adoption of healthier habits.[40] For the purposes of this trial, the personalised group Do's (from ONMI) are based on subject's reports from the behavioural questionnaire at V2 in addition to inputs from the nutritional recommendations (from the Eurecat Nutrition Team)

**Table 3** Recommended foods and functional ingredients for each metabolic cluster

| | |
|---|---|
| Carbohydrate | Functional ingredient: *FOS and †Inulin<br>Prebiotics: fibre-rich plants (Jerusalem artichoke, onion, leek, asparagus, kale) |
| Microbiota | Functional ingredient: *FOS and †Inulin<br>Prebiotics: fibre-rich plants (Jerusalem artichoke, onion, leek, asparagus, kale)<br>Fermented vegetables<br>Vegetables rich in fibre |
| Lipid | Functional ingredient: sunflower oil<br>Raw nuts and seeds<br>Omega 3 and 6: chia seeds, hemp seeds, walnuts, flax seeds<br>Vegetables rich in fibre |
| Inflammation | Functional ingredient: turmeric powder<br>Raw nuts and seeds<br>Omega 3 and 6: chia seeds, hemp seeds, walnuts, flax seeds<br>Dark chocolate |
| Oxidative stress | Functional ingredient: oleic acid enriched sunflower oil.<br>Raw nuts and seeds<br>Orange, yellow, red coloured vegetables (rich in vitamin A, C, E)<br>Dark chocolate<br>Vegetables rich in fibre |

*FOS originates from partial hydrolysis of chicory roots.
†Inulin is extracted from chicory roots.
FOS, fructooligosaccharide.

via the PREVENTOMICS platform, to provide a comprehensive behavioural change and improve adherence to the dietary intervention. For example, if a participant was recommended to eat kale and brussels sprouts, they could get a message like: 'Our analysis shows kale and brussels sprouts are good for you and should be part of your diet. Find out how much you should be consuming. Do it now'. Table 4 illustrates the different types and quantity of the Do's and massages delivered to personalised and control groups. The messages delivered to the control group are not personalised and are mostly informational in nature rather than prompting participants to take a specific action (ie, general guidelines available from the National Health Service and the WHO). The personalised and control groups receive the same behavioural treatment in terms of volume (frequency and intensity); the content of messages differs between groups as reflected in the numbers of specific types of messages delivered to each group, but the total number of messages is very similar (table 4).

### Compliance and food intake biomarkers
Dietary adherence is assessed twice a week—through an electronic questionnaire—by reporting the proportion of food consumed from the meals provided by Simple Feast in the previous 3 days. For example, in response to the question 'How much of your breakfast did you eat on day 1?' the possible answers can be: (1) Nothing or very little (0%–30%); (2) Approximately half (30%–70%); (3) Almost everything or everything (70%–100%).

Overall compliance to the diet and the behavioural programme is measured at the end of the trial by a six-point Likert scale question ranging from 1 (not at all compliant) to 6 (completely compliant). In addition, objective measures of adherence to the nutrition intervention will be evaluated: urine collected at prebaseline and the end-of-trial visits will be analysed for the quantification of selected biomarkers of food intake through a target ultra-performance liquid chromatography-ion mobility separation-high resolution mass spectrometry approach at the University of Parma (UNIPR, Italy). In particular, a set of about 150 potential biomarkers of intake related to major food groups (tubers, cereals, legumes, vegetables, fruits, nuts, vegetable oils, dairy products, meat, fish and alcoholic beverages; among others) and, when available, specific foodstuffs (orange, apple, cocoa, etc) will be defined on the basis of available literature and the technical feasibility at the time of the sample analysis. Data on biomarkers of food intake will also serve to assess the validity of the information collected through 3-day food diaries at prebaseline and end-of-trial visits.

### Data collection and outcome measures
Data are collected at screening, prebaseline (V1), baseline (V2) and 10 weeks after initiating the intervention (V3) by using self-reported questionnaires, biological specimens, accelerometers and physical examinations conducted by trained local research staff in accordance with standard operating procedures (table 5). All primary and secondary endpoints are derived from measurements obtained at baseline (V2) and at the end of the intervention (V3). Measurements taken prebaseline (V1) were used for clustering assignment. Reminder text messages are being sent to the participants before visit days. It is expected that a high retention rate will be achieved due to the nature of the study.

### Primary outcome
The primary endpoint is the difference in body fat mass (kg) from baseline to end of trial between the two intervention groups (personalised vs control). Body fat mass is determined during body composition analysis by use of dual-energy X-ray absorptiometry (GE Healthcare Lunar, Madison, Wisconsin, USA). The participants are scanned in a fasted state, lying on their back wearing lightweight clothes without jewellery and other metallic objects.

### Secondary outcomes
#### Anthropometry and body composition
Subjects' body composition is measured using a whole body DXA scan. Body weight is measured at all visits using a calibrated digital scale to the nearest 0.1 kg with participants wearing lightweight clothes and no shoes, as well as after voiding their bladder. Height is measured at the screening visit using a wall-mounted stadiometer to the

**Table 4** The type and number of behavioural messages delivered by ONMI to the participants in the personalised and control groups

| Type of messages | Quantity | Description | Example | PP | C |
|---|---|---|---|---|---|
| Starter Do | 1 | Easy start of the programme on behavioural questionnaire completion at V2 | SWITCH SEATS DAY! Move some seating around today. Sit somewhere different at meals/ when working/when watching TV. Get a new view!—Shaking up old habits is good for you and puts you back in charge of your life. Try something new regularly. Make every day count! -- | ✓ | ✓ |
| General Do | 5 | Apply to everyone, relatively easy, to get user hooked to the programme | NEW WAY DAY. Take a detour today, go the prettiest route not the shortest. Allow more time, smile at people. Spot 3 beautiful things along the way.—Wakey Wakey. Regularly challenging our brain keep us alert and interesting. When we take notice of our surroundings we start to live life to the fullest. | ✓ | |
| Personalised Do | 10 | Based on behavioural questionnaire | WHAT ARE YOU EATING FOR? Back off from boredom, address your stress. Get busy, unwind, release your emotions so you only eat when you're hungry today. | ✓ | |
| System Message | 3 | Encouragements, tips, manage expectations | HEALTH TIP. Regular contact with friends and family is key to good mental and physical health. Connections give meaning and purpose to our lives, even when it is digitally. | ✓ | |
| Expander Do | 3 | Prompt user to explore new parts of personality, based on behavioural questionnaire | EXPANDER: It's NO Day today. Don't say yes when you really want to say no. Give no reason or excuse. Just say, 'Sorry, but the answer's no'. | ✓ | |
| Preventomics Messages | 6 | Template messages that use inputs from the nutritional recommendations of food to increase | PREVENTOMICS: Are you getting the right amount of {{.R1}} and {{.R2}} in your diet? Go online and find some interesting recipes to try at home. Do it now. | ✓ | |
| General Messages | 24 | Recommendations from the NHS and WHO on eating, eating out, exercise, check-ups, help and support, balanced diet | Eating a healthy, balanced diet is an important part of maintaining good health, and can help you feel your best. This means eating a wide variety of foods in the right proportions, and consuming the right amount of food and drink to achieve and maintain a healthy body weight. | | ✓ |

C, control; NHS, The National Health Service; PP, personalised plan; PREVENTOMICS, Empowering consumers to PREVENT diet-related diseases through OMICS sciences; WHO, World Health Organization.

nearest 0.5 cm while participants are not wearing shoes. BMI ($kg/m^2$) is calculated as weight in kilograms divided by height in metres squared. Waist circumference (cm) is measured with a stretch-resistant tape at the midpoint between the lower margin of the last palpable ribs and the top of the iliac crest, in the fasted state (non-fasted during the screening visit) with an empty bladder and with participants wearing light clothes. Each measurement is taken twice to the nearest 0.5 cm and the average is used.

## Biological samples

Fasting blood samples collected at prebaseline, baseline and week 10 are analysed for plasma glucose, insulin, adipokines (leptin and adiponectin), inflammatory biomarkers (CRP, IL-6, IL-10, TNF-α, MCP-1, sICAM-1, sCD-14), lipid profile (total cholesterol, LDL-cholesterol, HDL-cholesterol, oxLDL, triglycerides), liver biomarkers (ALT and GGT) and renal biomarkers (uric acid, creatinine). Blood samples are sent to Eurecat for metabolomics analysis while biochemical markers are measured at NEXS, University of Copenhagen.

## Blood pressure

Systolic and diastolic blood pressures and heart rate are measured by an automatic sphygmomanometer on the arm after 5–10 min rest in a sitting position. The same arm is used during all visits. The measurement is repeated three times (or four, if the last two measurements deviate by >5 mm Hg), approximately 1 min apart, and readings are recorded to the nearest 1 mm Hg for blood pressures and 1 bpm for heart rate. The average of the last two readings is used.

## Other outcome measures

### Saliva

Saliva samples were collected prebaseline and sent to Alimentomica (University of the Balearic Islands, Palma de Mallorca) for analysis of genetic variants, mainly SNPs in genes related to metabolism, inflammation, and oxidative stress for cluster assignment. Currently, there are 188 candidate SNPs being analysed and validated in an ongoing trial and a minimum of 35 and maximum of 150 SNPs of those are expected to be used in this study.

### Urine samples

Participants were asked to deliver a second-void urine spot sample in the morning at pre-baseline, and do the same at baseline and end of the study visits. Urine samples from prebaseline and week 10 are used by UNIPR for food intake biomarker analysis and by Eurecat for analysing markers of oxidative stress used in cluster assignment.

**Table 5** Procedures and activities during the study period

| | Screening visit (V0) | Pre-baseline visit (V1) | Baseline visit (V2) | End of trial visit (V3) |
|---|---|---|---|---|
| **Week** | −20 to −13 | −9 to −6 | 0 | 10 |
| **Visit day** | 0 | 1 | 2 | 3 |
| Informed consent | X | | | |
| Review of inclusion and exclusion criteria | X | | | |
| Medical history and examination | X | | | |
| Randomisation | | | X* | |
| Registration of medication and adverse events | X | X | X | X |
| **Anthropometry** | | | | |
| Body weight | X | X | X | X |
| Height | X | | | |
| Waist circumference | X | X | X | X |
| Body composition (DXA) | | | X | X |
| **Biological samples** | | | | |
| Fasting blood sample | | X | X | X |
| Saliva sample (SNPs) | | X | | |
| Urine sample | | X | X | X |
| Faecal sample | | | X† | X† |
| **Nutritional assessment** | | | | |
| Food Frequency Questionnaire | | X | | X |
| 3-day dietary records | | X† | | X† |
| **Other measurements** | | | | |
| Blood pressure/heart rate | | X | X | X |
| Three factor eating questionnaire | | | X | X |
| Perceived Stress Scale | | | X | X |
| Behavioural questionnaire (by ONMI) | | | X | X |
| Quality of Life questionnaires | | | X | X |
| Diet satisfaction questionnaire (DSat-28) | | | X | X |
| Money spent on food questions | | | X | X |
| Accelerometer (sleep and PA) | | X‡ | X‡ | |

*Prior to visit.
†At home activity prior to visit.
‡At home activity following the visit.
DXA, dual-energy X-ray absorptiometry; PA, physical activity.

Furthermore, aliquots of urine samples from all visits are stored in a biobank for future analyses.

### Nutritional assessment

Dietary intake is assessed pre-baseline and at the end of the study by using a validated self-administered electronic form of the European Prospective Investigation of Cancer-Norfolk Study food frequency questionnaire,[41] supervised by a trained staff member. Furthermore, participants are instructed to complete 3-day weighed food records during the week before the pre-baseline visit and during the third week of the intervention. The dietary records cover two non-consecutive weekdays and 1 weekend day. Nutrient analysis will be done by the software program Vitakost (Conava ApS; Kolding, Denmark), which is based on the Danish national food database.

### Questionnaires

**Eating behaviour** assessment at baseline and week 10 is conducted by administrating the three factor-eating questionnaire.[42] This is a 51-item self-report questionnaire which measures three domains of eating behaviour: (1) cognitive restraint of eating, (2) disinhibition and (3) hunger.

**Stress** assessment is conducted through the 10-item perceived stress scale (PSS)[43] at baseline and week 10. The PSS is one of the most widely used psychological instruments. It measures the degree to which participants

perceive events in their life as being stressful by asking about thoughts and feelings over the last month using a response scale from 0 (never) to 4 (very often).

**Quality of life** is assessed by two questionnaires at baseline and week 10: (1) The EuroQol five-dimension five-level (EQ-5D-5L),[44] which is a standardised instrument developed by EuroQol Group for measuring health-related quality of life on five dimensions (mobility, self-care, usual activities, pain/discomfort and anxiety/depression), with five response levels per dimension, and also includes a Visual Analogue Scale (EQ) by which respondents report their perceived health status and (2) Obesity and Weight-Loss Quality of Life Instrument,[45] which is an instrument consisting of 17 statements about weight-related feelings and emotions which are rated on a seven-point scale, and primarily measures emotions and feelings resulting from suffering from obesity and trying to lose weight.

**Diet satisfaction** is assessed at baseline and week 10 by the Diet Satisfaction Questionnaire (DSat-28)[46] that involves 28 statements grouped into five dimensions (healthy lifestyle, eating out, cost, preoccupation with food and planning and preparation) to evaluate satisfaction with weight-management diets.

**Food expenditure** is assessed by two questions completed at baseline and week 10 regarding the amount of money spent on food for the whole household.

### Physical activity and sleep

Physical activity and sleep patterns are determined by ActiGraph GT3X+accelerometer (ActiGraph, Pensacola, Florida, USA) for 7 days/8 nights immediately following the prebaseline visit and again for 7 days/8 nights during the third week of the intervention. During these wear-periods, a self-administered sleep-log to assess bedtimes is also obtained.

### Gut microbiota analyses

Microbiota composition will be determined on faecal samples collected within 24 hours prior to the clinical investigation days at baseline and week 10. These samples are not used in the cluster assignment but are collected and stored for future analyses.

### Data management

Study investigators have access to the data collection forms and protocols using a secure shared drive. All collected data are pseudoanonymised and participants are identified only by a study ID number on documents and on electronic databases, with personal identifiers kept separately under strict access control, limited to investigators and study staff directly involved in data collection and entry. All biological specimens sent to the consortium partners are pseudoanonymised and encrypted. On completion of the study, data will be stored in a password-protected database, accessible only by study investigators, in anonymous form for a minimum of 10 years.

### Data monitoring

As the intervention risks to participants are minimal, a Data and Safety Monitoring Board was not deemed necessary. However, in case of unexpected adverse events during the study period, these are recorded and brought up to the principal investigator for appropriate decision-making. We did not plan to conduct interim analysis for safety as we did not anticipate any serious adverse events that would require trial termination.

### Power and sample size calculation

To detect a difference in body fat mass change of 1.25 kg between the two intervention groups with 80% power at a two-tailed level of significance of 0.05, assuming a SD of 2.0 kg, a sample size of 41 per group (ie, personalised vs control) is needed, that is, a total sample size of 82 completers. To allow for an anticipated 18% dropout rate, 50 subjects per group would need to be recruited (total n=100). The expected difference in fat mass between groups (1.25 kg) and the associated SD (2.0 kg) were based on values calculated from the raw data of the SHOPUS study.[47] In that study, we reported on body weight and fat mass during a 6-month dietary intervention in subjects with BMI of 22.6–47.3 kg/m$^2$. To conduct power calculations for this study, we extracted raw data from the SHOPUS study that were most representative of the current subjects and intervention duration. Accordingly, we selected those SHOPUS participants with BMI ≥27 kg/m$^2$ and elevated waist circumference (males >94 cm; females>80 cm) which represented more than 67% of the total study population (n=145) and assessed their body weight and fat mass at 12 weeks; this was an interim time point of the SHOPUS study (not published in the original paper),[47] which was the closest to the 10-week time frame of the current intervention.

### Statistical analysis

Data analysis will be conducted using IBM SPSS Statistics version 28 (IBM Corp., Armonk, NY, USA) and R. Before statistical analyses are conducted, all continuous variables will be tested for normality and homogeneity of variance. For our primary objective, differences in fat mass from baseline to end of trial (V3 minus V2) between the two intervention groups (personalised vs control) will be compared by means of linear mixed models with time and randomisation group as main effects, a time-by-group interaction, and adjusting for potential confounders (eg, sex, age and BMI at baseline) as necessary. If significant interactions emerge, post hoc testing will be used to evaluate effects within the metabolic clusters. LMMs are able to handle possible imbalances between groups in case of missing values in a single response variable.

### ETHICS AND DISSEMINATION

The study has been approved by the Regional Committees on Health Research Ethics, Region Hovedstaden in Denmark (H-20029882) and is being conducted in

accordance with the Helsinki Declaration. Any protocol amendments are submitted to the Research Ethics Board for approval and communicated to study participants and the trial registry once approved.

All personal data is being handled confidentially and stored in accordance with applicable law, GDPR and Danish Data Protection Agency. Participants received written and oral information on the study procedures, and only trained study-personnel provided information, monitored and attested signing of the informed consent form (online supplemental material 3). In addition, an optional GDPR consent to provide excess sample materials to study biobank was signed. The Research biobank has been approved by the Danish Data Protection Agency.

A manuscript with the results of the primary study will be submitted for publication to an international, peer-reviewed journal, regardless of whether results are positive, negative or inconclusive in relation to the study hypothesis. Authorship eligibility will be based on the recommendations from the International Committee of Medical Journal Editors. On completion of the trial, and after publication of the primary manuscript, data requests can be submitted to the principal investigator at the Department of NEXS at the University of Copenhagen, Denmark.

## Perspective

The results from this study will serve as proof of concept for the efficacy of using metabolic and genetic biomarkers to provide personalised diet treatments for reducing body fat mass and subsequently for improving health outcomes, such as metabolic and inflammatory markers, in high-risk individuals. Moreover, the findings will inform recommendations regarding the efficacy of using a web-based DSS for personalising dietary plans to support and maintain health-promoting behaviours.

**Author affiliations**
[1]Department of Nutrition, Exercise and Sports, Faculty of Science, University of Copenhagen, Copenhagen, Denmark
[2]Department of Clinical Nutrition, College of Applied Medical Sciences, King Saud bin Abdulaziz University for Health Sciences, Riyadh, Saudi Arabia
[3]R&D, Food and Culinary Department, Simple Feast, Copenhagen, Denmark
[4]Laboratory of Molecular Biology, Nutrition and Biotechnology - NUO group, University of the Balearic Islands, Palma, Spain
[5]Spin-off n.1 of the University of the Balearic Islands, Alimentómica S.L, Palma, Spain
[6]Human Nutrition Unit, Department of Food and Drug, University of Parma, Parma, Italy
[7]ONMI, Eindhoven, The Netherlands
[8]R&D, Food Division, Grupo Carinsa, Barcelona, Spain
[9]Biotechnology Area, Nutrition and Health Unit, Eurecat Centre Tecnològic de Catalunya, Reus, Spain
[10]Healthy Weight Center, Novo Nordisk Foundation, Hellerup, Denmark

**Contributors** The overall framework of the EU-project PREVENTOMICS was initiated by BG, AC and JDB. The overall design of the present Danish study involved all coauthors. Detailed planning, implementation and daily management of the Danish study is carried out by KP, MAA, FM and MFH. MAA, FM and MFH drafted the initial manuscript. KP, SMOG, PM, FS, MP, MW, AC, JDB and BG critically reviewed and edited the manuscript. All authors approved the final version for publication.

**Funding** This work and the European partners in the PREVENTOMICS project are supported by the European Union's Horizon 2020 research and innovation programme under grant agreement No. 818318.

**Competing interests** None declared.

**Patient and public involvement** Patients and/or the public were not involved in the design, or conduct, or reporting, or dissemination plans of this research.

**Patient consent for publication** Not applicable.

**Provenance and peer review** Not commissioned; externally peer reviewed.

**ORCID iDs**
Mona Adnan Aldubayan http://orcid.org/0000-0002-5398-6673
Kristina Pigsborg http://orcid.org/0000-0003-1987-523X

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
