## [Reviewer comments · BMJ Open]

ARTICLE DETAILS

TITLE (PROVISIONAL)	Empowering consumers to PREVENT diet-related diseases through OMICS sciences (PREVENTOMICS): Protocol for a parallel double-blinded randomised intervention trial to investigate biomarker-based nutrition plans for weight loss
AUTHORS	Aldubayan, Mona A.; Pigsborg, Kristina; Gormsen, Sophia; Serra, Francisca; Palou, Mariona; Mena, Pedro; Wetzels, Mart; Calleja, Alberto; Caimari, Antoni; Del Bas, Josep; Gutierrez, Biotza; Magkos, Faidon; Hjorth, Mads

VERSION 1 – REVIEW

REVIEWER	Koutoukidis, Dimitrios University of Oxford, Nuffield Department of Primary Care Health Sciences
REVIEW RETURNED	09-Jun-2021

GENERAL COMMENTS	This protocol presents an interesting and provocative trial. Before publication, substantial changes are needed to allow readers to understand and be able to reproduce the intervention. 1. Please add substantial details on the meals that each group will receive. Since participants are provided with the foods, as an absolute minimum, this should include the detailed nutritional information (macronutrients + amounts of each bioactive ingredient) of all the exact meals of each cluster and control group for at least the first 2 weeks. It is not sufficient to say that they foods will have turmeric or chia seeds. The quantities matter if you were to examine the intervention efficacy. You can add all this in the supplementary information. Ideally, information about all the provided meals should be added.2. It is unclear why participants receive 12 meals a week (2 per day) and not 14. What are they advised to have the 7th day?3. What advice do they receive for the rest of the meals that are not provided? Specific details are needed.4. I cannot see how a 2000 kcal diet for women and a 2500 kcal diet for men would lead to substantial weight loss. This is important and worth discussing the rationale further in the manuscript because weight / fat mass loss is the main aim of the study but the authors aim to achieve this only in a subsample (i.e. those w obesity).5. Much more detail is needed on the description of the behavioural intervention. I suggest that the authors use the TIDIER checklist for this to specify the frequency, duration, and intensity of support in detail. Also they should report on the behaviour change techniques to be used and specify how these would differ between groups as there is a hint in the manuscript that the techniques and the intensity of support might differ
---

	between groups. If so, we can reasonably assume that this may lead to more weight loss to the group with the more support based on previous literature. 6. Unfortunately, I strongly question the sample size calculation. The author expect an SD of 2 in fat mass and cite a previous article. However, in the cited paper, the SD of fat mass is 10 at baseline and the SD of change in fat mass is 22. Such larger numbers are to be expected because all weight loss trials have substantial variation in SD of weight and fat mass typically >7. However, I do see this as a proof of concept trial but this would mean focusing on process outcomes such as feasibility, engagement, and dietary adherence. 7. Details of the specific FFQ is needed, on the process of the 3-day diet records, and on the software and personnel to be used for analysis. 8. The authors claim that a big advantage of the trial is its double-blinded fashion. I think this holds promise, but details on this need to be added. It is unclear how both the participants and the researchers are going to be blinded and how this will be maintained throughout the intervention period. 9. A few typographical and grammatical errors to be corrected prior to publication. 10. More details on the stats are needed. Suggest that the authors state where they will publish their detailed statistical analysis plan ahead of database lock.
--	--

REVIEWER	Zaslavsky, Oleg University of Washington
REVIEW RETURNED	15-Jun-2021

GENERAL COMMENTS	Thank you for the opportunity to review the study protocol by Aldubayan and colleagues that aims at determining the effects of precision diet on body composition measures among adults with central obesity. This is an important study that should yield interesting results. The protocol is well described but there are multiple gaps that need to be addressed. I list them in the order they appear in the text. PP. 10. More details are needed about approaches used to calculate metabolic clusters. Even though the authors acknowledged that specific algorithms cannot be disclosed due to IP issues, there are still opportunities to describe the overall approach in more depth. It is unclear how scores are calculated. Will they consider each marker separately and calculate scores based on the observed distribution in the sample or based on pre-defined cutpoints? What about individuals with deficits across domains? How priority scores are assigned? How they will handle metabolically healthy individuals? I believe they should be able to give general outlines without giving away their 'secret sauce' PP. 11 The part about 'bioactive compound' is very surprising. There was no any notion about the compound earlier in the text. It seems that in addition to personalized foods, this study is also about bioactive ingredients. Would it be a fair statement? Can the authors explain what role the bioactive compound plays? Are there specific hypotheses concerning the compound? Will people in the control group receive the bioactive compound? PP. 12 There is a notion of push notifications. How many notifications will be sent?
--

	Are participants expected to respond to the notifications? What does it mean notification in the control group will not be based on the same behavioral change methodology? So what methodology will they use? Will they receive the same number of push notifications? Will they use the same method of receiving and responding to notifications? What does compliance with Do messages mean? Do participants expect to report if they took an action after receiving a message? When are urine samples to monitor adherence collected? Will the information on adherence used to adjust foods/messages? PP. 18 The authors described power calculations based on a previous study in a comparable population and provided a reference for the study. But when one examines the study the details are different from what is stated in the text. For example, intervention time in the referenced study was 6 months and not 12 weeks. The observed difference in body fat between intervention and control arms at the end of 6 months fat was 2.7 (3.9-1.5) and not 1.25 kg.
--	--

VERSION 1 – AUTHOR RESPONSE

Reviewer: 1

Dr. Dimitrios Koutoukidis, University of Oxford Comments to the Author:

This protocol presents an interesting and provocative trial. Before publication, substantial changes are needed to allow readers to understand and be able to reproduce the intervention.

Reply: Thank you. We have revised our manuscript accordingly. Following please find our point-by-point responses to your comments; the corresponding revisions in the manuscript are highlighted in red font.

1. Please add substantial details on the meals that each group will receive. Since participants are provided with the foods, as an absolute minimum, this should include the detailed nutritional information (macronutrients + amounts of each bioactive ingredient) of all the exact meals of each cluster and control group for at least the first 2 weeks. It is not sufficient to say that they foods will have turmeric or chia seeds. The quantities matter if you were to examine the intervention efficacy. You can add all this in the supplementary information. Ideally, information about all the provided meals should be added.

Reply: We have added quantitative information on the target daily amounts of energy and macronutrients for the provided meals, together with the various bioactive ingredients for each diet cluster, as well as an example of a 3-day menu as a supplementary file (see supplementary material 2). The corresponding text has been amended accordingly (lines 208-218).

2. It is unclear why participants receive 12 meals a week (2 per day) and not 14. What are they advised to have the 7th day?

Reply: Indeed, participants were provided with 12 meals/week. This was because of a combination of factors including budgetary limitations (available funding did not allow for more meals/week), practical reasons (fixed delivery schedule of fresh meals twice/week would need to increase), and behavioral reasons (allow subjects a 'free' Saturday). We have clarified these details in the manuscript (lines 189-191).

3. What advice do they receive for the rest of the meals that are not provided? Specific details are

needed.

Reply: For the days for which meals were not provided (Saturdays), as well as for all lunches, participants were encouraged to refer to the recipe recommendations that were given to them through the App, so they prepare meals as similar as possible to those to which they were assigned. We have updated the manuscript to reflect this (lines 191-194).

4. I cannot see how a 2000 kcal diet for women and a 2500 kcal diet for men would lead to substantial weight loss. This is important and worth discussing the rationale further in the manuscript because weight / fat mass loss is the main aim of the study but the authors aim to achieve this only in a subsample (i.e. those w obesity).

Reply: We used the average population requirements of 2000 kcal/day for women and 2500 kcal/day for men as a rule of thumb for calculating the energy content of breakfast and dinner (we did not provide lunch). Also, participants were instructed to eat until they were comfortably full (ad libitum). We anticipated that the personalized diets would result in favorable changes in eating behavior and/or physiological and metabolic parameters that would promote body weight and fat loss. The manuscript has been revised to reflect this (lines 195-204).

5. Much more detail is needed on the description of the behavioural intervention. I suggest that the authors use the TIDIER checklist for this to specify the frequency, duration, and intensity of support in detail. Also, they should report on the behaviour change techniques to be used and specify how these would differ between groups as there is a hint in the manuscript that the techniques and the intensity of support might differ between groups. If so, we can reasonably assume that this may lead to more weight loss to the group with the more support based on previous literature.

Reply: We apologize for this misunderstanding. Both groups (personalized and control) received similar attention to the behavioral treatment in terms of frequency and intensity. Only the content of the messages and prompts differed. Accordingly, amendments were made to the manuscript to clarify these issues and some examples of the messages for the two groups have been included in table 4 (lines 236-239, page 12).

6. Unfortunately, I strongly question the sample size calculation. The author expect an SD of 2 in fat mass and cite a previous article. However, in the cited paper, the SD of fat mass is 10 at baseline and the SD of change in fat mass is 22. Such larger numbers are to be expected because all weight loss trials have substantial variation in SD of weight and fat mass typically >7. However, I do see this as a proof of concept trial but this would mean focusing on process outcomes such as feasibility, engagement, and dietary adherence.

Reply: We apologize for this. To get more robust estimates of sample size, we pulled raw data from the study described in the cited paper that were more representative of the subjects and intervention duration of the current study. The cited paper reported on a 6-month intervention among people with BMI 22.6-47.3 kg/m². We thus selected those subjects with BMI ≥ 27 and elevated waist circumference and used interim data at 12 weeks (closest time point to the current 10-week intervention) rather than 6 months. These data informed on a difference between groups of 1.25 kg with an SD of 2 kg (these numbers do not appear in the cited paper) that we used for power calculations in this study. Amendments were made accordingly in the power and sample size section (lines 358-365).

7. Details of the specific FFQ is needed, on the process of the 3-day diet records, and on the software and personnel to be used for analysis.

Reply: These details have been added to the revised manuscript (lines 304-310).

8. The authors claim that a big advantage of the trial is its double-blinded fashion. I think this holds promise, but details on this need to be added. It is unclear how both the participants and the researchers are going to be blinded and how this will be maintained throughout the intervention

period.

Reply: We have included more information on how the blindness for both the participants and the researchers will be maintained throughout the study (172-179 and 187-188).

9. A few typographical and grammatical errors to be corrected prior to publication.

Reply: We apologize for these errors. We have scrutinized the manuscript to correct all typos and grammatical mistakes.

10. More details on the stats are needed. Suggest that the authors state where they will publish their detailed statistical analysis plan of database lock.

Reply: We have amended the statistical analysis section to provide some of these details (lines 368-374). We do not intend to publish a detailed statistical analysis plan ahead of database lock, other than the one already publicly available at the clinical trial registry.

Reviewer: 2

Dr. Oleg Zaslavsky, University of Washington Comments to the Author:

Thank you for the opportunity to review the study protocol by Aldubayan and colleagues that aims at determining the effects of precision diet on body composition measures among adults with central obesity. This is an important study that should yield interesting results. The protocol is well described but there are multiple gaps that need to be addressed. I list them in the order they appear in the text.

Reply: Thank you. We have revised our manuscript accordingly. Following please find our point-by-point responses to your comments; the corresponding revisions in the manuscript are highlighted in red font.

PP. 10. More details are needed about approaches used to calculate metabolic clusters. Even though the authors acknowledged that specific algorithms cannot be disclosed due to IP issues, there are still opportunities to describe the overall approach in more depth. It is unclear how scores are calculated. Will they consider each marker separately and calculate scores based on the observed distribution in the sample or based on pre-defined cutpoints? What about individuals with deficits across domains? How priority scores are assigned? How they will handle metabolically healthy individuals? I believe they should be able to give general outlines without giving away their 'secret sauce'

Reply: We have added more information in the revised manuscript about the algorithm and the process used to assign participants into clusters (lines 158-165).

PP. 11 The part about 'bioactive compound' is very surprising. There was no any notion about the compound earlier in the text. It seems that in addition to personalized foods, this study is also about bioactive ingredients. Would it be a fair statement? Can the authors explain what role the bioactive compound plays? Are there specific hypotheses concerning the compound? Will people in the control group receive the bioactive compound?

Reply: The 'bioactive compounds' are simply other nutrients naturally present in personalized food items. As an example, the carbohydrate and microbiota cluster will receive fructooligosaccharide and inulin as bioactive ingredients; the lipid cluster will receive sunflower oil; the inflammation cluster will receive turmeric powder; and the oxidative cluster will receive oleic acid enriched sunflower oil. These bioactive compounds were selected following review of the literature and they are believed to benefit especially (or exclusively) the metabolic function of those in the corresponding diet cluster. We revised the manuscript to clarify this (lines 208-218).

PP. 12 There is a notion of push notifications. How many notifications will be sent? Are participants expected to respond to the notifications? What does it mean notification in the control group will not be based on the same behavioral change methodology? So what methodology will they use? Will they receive the same number of push notifications? Will they use the same method of receiving and

responding to notifications? *What does compliance with Do messages mean? Do participants expect to report if they took an action after receiving a message? When are urine samples to monitor adherence collected? Will the information on adherence used to adjust foods/messages?

Reply: We have revised the behavioral and compliance sections of the manuscript (lines 219-239 and table 4 in pages 12-13) to clarify all this information. Both groups (control vs. personalized) will receive similar attention via the same methodology (push notifications and messages via the App). The push notifications will only differ in the content, in that the prompts in the personalized group are specific to their diet cluster and their individual behavior traits (which were assessed at baseline), whereas the prompts in the control group are generic messages with guidelines from the WHO/NHS. Both groups will be required to respond to some actionable prompts (“Do”) by clicking on a button that confirms they read the message. However, these responses will not be tracked as a measure of compliance. Overall compliance to the intervention (consumption of meals and behavioral program) will be measured at the end of the trial by a Likert-scale question, and also by measuring biomarker levels in urine, but the messages delivered will not be adjusted according to this information (see also lines 240-253).

PP. 18 The authors described power calculations based on a previous study in a comparable population and provided a reference for the study. But when one examines the study the details are different from what is stated in the text. For example, intervention time in the referenced study was 6 months and not 12 weeks. The observed difference in body fat between intervention and control arms at the end of 6 months fat was 2.7 (3.9-1.5) and not 1.25 kg.

Reply: We apologize for this. To get more robust estimates of sample size, we pulled raw data from the study described in the cited paper that were more representative of the subjects and intervention duration of the current study. The cited paper reported on a 6-month intervention among people with BMI 22.6-47.3 kg/m². We thus selected those subjects with BMI ≥27 and elevated waist circumference and used interim data at 12 weeks (closest time point to the current 10-week intervention) rather than 6 months. These data informed on a difference between groups of 1.25 kg with an SD of 2 kg (these numbers do not appear in the cited paper) that we used for power calculations in this study. Amendments were made accordingly in the power and sample size section (lines 358-365).

VERSION 2 – REVIEW

REVIEWER	Koutoukidis, Dimitrios University of Oxford, Nuffield Department of Primary Care Health Sciences
REVIEW RETURNED	14-Nov-2021

GENERAL COMMENTS	Thank you for addressing our comments. A few additional essential points before publication are highlighted below:  - Line 170: Computer-generated randomisation: More details needed .e.g. with what method exactly? block randomisation? Substantial details should be added. - Lines 172-179: blinding. It is quite confusing how the details of the group and cluster of each participant will be passed on to a NEXS person and will be available in the platform but both the participants and local staff will be blinded. Please consider providing sufficient additional details, so that readers can reproduce the process. - Lines 199-202: The rationale about lower consumption at lunch / snacks due to the high-calorie plant-based breakfast/dinner needs to be supported by some literature. What recipe ideas do they get
---

	for lunch and is there a recommended or expected calorie limit/goal. If not too late, I would suggest that you advise them to eat as little as possible throughout the rest of the day (e.g. eat veggies/salads, drink energy-free drinks etc), so that they achieve the desired energy deficit. The concept of "ad libitum" is challenging for breakfast and lunch. I understand what you mean (that they should only eat as much as they want from the meal) but typically the concept of ad lib applies to studies where the food is abundant. So what if they actually want to eat more than the provided food? Are they allowed or encouraged? (I assume not). A better term is needed here, but I am not sure what the right term is. Line 249: "selected biomarkers of food intake". Please specify. Add randomisation in the list of activities in table 5. Sample size calculation: So the SD of change in fat mass was 2kg in the subsample of the SHOPUS trial? Worth clarifying. How many people were in this subsample? I will take your word for an expected SD of 2kg but I would strongly caution that this might not be achievable and you would probably need to revise your sample size calculation, expected effect size, or your primary outcome. Line 371: Why would you adjust for sex and age? these should be taken care of due to randomisation. Line 371: Why do you need a time-by-group interaction? This is only pre-post and due to randomisation, the baseline fat mass should be the same between groups.
--	--

REVIEWER	Zaslavsky, Oleg University of Washington
REVIEW RETURNED	02-Nov-2021

GENERAL COMMENTS	The authors were responsive to the reviewer's concerns and suggestions. This reviewer does not have any further comments
--